# Recent Advances in Using Natural Antibacterial Additives in Bioactive Wound Dressings

**DOI:** 10.3390/pharmaceutics15020644

**Published:** 2023-02-14

**Authors:** Meysam Firoozbahr, Peter Kingshott, Enzo A. Palombo, Bita Zaferanloo

**Affiliations:** 1Department of Chemistry and Biotechnology, School of Science, Computing and Engineering Technologies, Swinburne University of Technology, Hawthorn, VIC 3122, Australia; 2ARC Training Centre Training Centre in Surface Engineering for Advanced Materials (SEAM), School of Engineering, Swinburne University of Technology, Hawthorn, VIC 3122, Australia

**Keywords:** antibacterial additives, natural products, polymer wound dressing, endophytic fungi

## Abstract

Wound care is a global health issue with a financial burden of up to US $96.8 billion annually in the USA alone. Chronic non-healing wounds which show delayed and incomplete healing are especially problematic. Although there are more than 3000 dressing types in the wound management market, new developments in more efficient wound dressings will require innovative approaches such as embedding antibacterial additives into wound-dressing materials. The lack of novel antibacterial agents and the misuse of current antibiotics have caused an increase in antimicrobial resistance (AMR) which is estimated to cause 10 million deaths by 2050 worldwide. These ongoing challenges clearly indicate an urgent need for developing new antibacterial additives in wound dressings targeting microbial pathogens. Natural products and their derivatives have long been a significant source of pharmaceuticals against AMR. Scrutinising the data of newly approved drugs has identified plants as one of the biggest and most important sources in the development of novel antibacterial drugs. Some of the plant-based antibacterial additives, such as essential oils and plant extracts, have been previously used in wound dressings; however, there is another source of plant-derived antibacterial additives, i.e., those produced by symbiotic endophytic fungi, that show great potential in wound dressing applications. Endophytes represent a novel, natural, and sustainable source of bioactive compounds for therapeutic applications, including as efficient antibacterial additives for chronic wound dressings. This review examines and appraises recent developments in bioactive wound dressings that incorporate natural products as antibacterial agents as well as advances in endophyte research that show great potential in treating chronic wounds.

## 1. Introduction

Wound care is a global health issue. A retrospective analysis of the Medicare dataset for 2014 in the USA indicated that 8.2 million Medicare beneficiaries had at least one type of wound or related infection [1], indicating that the financial burden equates to between US $28.1 billion to US $96.8 billion, including the cost of infection management [2].

Wound repair mechanisms consist of four main phases including haemostasis, inflammation, proliferation, and dermal remodelling. In the haemostasis phase, a blood clot is formed to prevent exsanguination from vascular damage. In this step, platelet receptors interact with extracellular matrix proteins to promote adherence to the blood vessel wall. The second phase of wound healing is inflammation which is the primary defence against pathogenic wound invasion, followed by proliferation as the third phase. In this phase, activation of keratinocytes, fibroblasts, macrophages, and endothelial cells will help the process of wound closure, matrix deposition, and angiogenesis. Finally, in the matrix remodelling phase, a fibrin clot is deposited leading to the formation of a scar. For more information, readers can refer to review papers by Wilkinson et al. [3] and Carr et al. [4], scrutinising various mechanisms of wound healing.

There are two classes of wounds—acute and chronic. Acute wounds are injuries to the skin which are healed by the normal process of wound repair [5]. Wounds that have not progressed through the normal repair process and remain unhealed for an extended period are referred to as chronic wounds. The latter type of wound is a burden to the healthcare system [6], to the extent that 2% of the national health expenditure in Australia is spent on these types of wounds, equivalent to more than AUD 3.5 billion annually [7].

The delayed and incomplete healing process of chronic non-healing wounds exposes patients to a high risk of infection. Thus, for severe and chronic wounds, more advanced treatments and wound dressings should be applied to assist with accelerating wound healing [8] and preventing infection [9].

Among the more than 3000 wound dressing types on the wound management market, different characteristics can be achieved based on the intrinsic properties of the polymers used in wound dressing preparation. These characteristics include their ability to absorb exudate, combat infection, relieve pain, promote autolytic debridement, or even provide and maintain a moist environment at the wound surface. However, there is no wound dressing that possesses all these properties. The type of wound dressing is selected based on the patient’s health status, wound type, location, depth, amount of exudate, wound adhesion, and economic considerations [10,11]. Hydrogels, foams, dermal patches, films, nanoparticles, hydrocolloids, nanofibers, and membranes are the main groups of dressings, and their description, characteristics, and polymers used to make them are summarised in Table 1 [10].

Studies show that the challenges of wound dressings linked to wound infection are significant. In most acute and chronic infections, a mixed population of both aerobic and anaerobic microorganisms is observed [62] and yet to be eliminated. This challenge emphasises the importance of strategies that target the most common bacteria on the wound surface. Data from recent studies on various wound infections (e.g., surgical incisions, burns, abscesses, and traumatic wounds) confirm the presence of *Pseudomonas* sp. and *Staphylococcus aureus* as the most common Gram-negative and Gram-positive bacteria, respectively, on wound surfaces with a share of 58.4% for Gram-negative and 41.6% for Gram-positive bacteria [63]. The percentage of each group of bacteria can be observed in detail in Figure 1.

There is a significant need for antibacterial wound dressings for controlling and reducing bacterial infections. One of the best approaches is wound dressings containing antibiotics as an additive. Numerous reports can be found on wound dressings containing antibiotic additives and their effects on wound dressings [64,65,66].

By using an efficient amount of antibiotics, suitable treatment of wound infections can be achieved. However, high amounts of antibiotics will cause systemic toxicity [67]. In order to overcome these detrimental effects, the antibacterial compounds and antibiotics are embedded in wound dressings for sustained and controlled drug release [10]. A lack of new antibiotics and antibacterial agents, as well as widespread distribution and misuse of these antibacterial compounds, has caused an increase in antimicrobial resistance (AMR). It has been proposed that AMR has the potential to kill ten million lives by 2050 worldwide, costing an estimated US $100 trillion [68]. This can result in the return to a pre-antibiotic era, with infections caused by multiple-resistant pathogens [31]. Thus, there is an urgent need for sustainable novel antibacterial additives to overcome this major clinical problem.

Looking at the newly approved drugs between 1981 and 2019, followed by the share of each source in previous researched studies, indicates that the share of natural or naturally inspired approved drugs has increased over time [69,70]. One of the most important sources of these novel pharmaceuticals is plants, which form a large portion of these newly approved drugs. Botanical-based natural antibacterial compounds have attracted great attention in recent years, indicating their great potential to be used as antibacterial additives in different medical applications including wound dressings.

There has always been a significant need for novel antibacterial additives to improve wound dressing characteristics. After investigating the importance of botanical-based antibacterial compounds as one of the most important sources of pharmaceuticals, the efficiency of the resultant developed dressings using these additives has been scrutinised in each of the main groups of polymer wound dressings. The focus of this review is on the dressings containing additives against targeted bacteria, including *P. aeruginosa*, *S. aureus*, *Escherichia coli,* and other wound-infecting bacteria.

## 2. Bioactive Wound Dressings (Polymer + Additives)

### 2.1. Hydrogels

Hydrogel wound dressings containing natural antibacterial bio-additives have been used extensively for wound treatments. These additives are able to optimise the antibacterial properties against the unfavourable increase of bacterial proliferation in hydrogel wound dressings [11,71].

Plant-based antibacterial additives have been used in hydrogels to increase their activity against bacteria in infected wounds [71]. One of the most important class of additives in this group are the essential oils. The presence of essential oils as hydrophobic compounds in the hydrogel texture leads to good mechanical properties, degradability, improvement of the porous structure, and antioxidant properties [72]. Altaf et al. used a solution casting method to produce a polyvinyl alcohol/starch hydrogel membrane containing various concentrations of clove essential oil. The products resulted in excellent antibacterial activity, with a minimum inhibition zone of 34 ± 0.42 mm against *S. aureus* and 31 mm against *E. coli* [71]. The synthesis scheme has been demonstrated in Figure 2. Other essential oils used in hydrogels are lavender and tea tree oil [73]. Using these two essential oils in gellan gum hydrogels at 25% *w*/*w* resulted in an efficient zone of inhibition of 20 mm against *S. aureus* and 30 mm against *E. coli* in standard disc diffusion assays [73]. Several studies have used different essential oils in hydrogels, such as basil oil [74], tea tree oil [75], sweet fennel oil [76], rosemary essential oil, orange essential oil [77], and *Thymus daenesis* oil [78], to improve their antibacterial activity.

In addition to essential oils, plant extracts have been used as additives in hydrogel wound dressings. In a study by Shukla et al., a bioactive hydrogel dressing containing an ethanolic extract of *Morus alba* leaves was used against diabetic wounds. The apigenin derived from the extract was tailored with gellan gum-poly ethylene glycol-chitosan hydrogels and screened in vivo for its effectiveness. The results indicated that the apigenin additives caused effective stimulation of wound contraction and increase in the collagen content in diabetic as well as normal wound tissues, which leads to an accelerated wound healing process [79]. The antibacterial activity of *Morus alba* extracts against *S. aureus* has been previously investigated, resulting in a minimum inhibitory concentration (MIC) of 250 µg/mL [80].

### 2.2. Hydrocolloids

Hydrocolloids have been previously used along with natural antibacterial additives to improve their characteristics against wound bacteria and reduce the unpleasant odour [81]. These additives are the extracts of some pre-approved antibacterial plants, such as *Centella asiatica* (CA) and *Phellodendri amurensis* (PA) [82,83], which have been used in different studies against several bacteria. After loading CA plant extracts in alginate hydrocolloids using a hot melting method, Jin et al. showed excellent swelling, drug release, and mechanical properties compared with similar commercial products. Enhanced healing process in excision, infection, and abrasion wounds were observed in a rat wound model, which suggests that this extract is a potential candidate for the treatment of various wounds [82]. The preparation technique has been demonstrated in Figure 3. Antibacterial activity tests of the CA extracts at 100 µg/mL against *P. aeruginosa*, *S. aureus*, and *E. coli* resulted in zones of inhibition between 28–30 mm [83].

Another application of hydrocolloids containing CA extracts is skin treatment. Kuo et al. produced an anti-acne patch with gelatin/chitosan (GC) bilayer hydrocolloid patches. This anti-acne bilayer patch was loaded with Cortex PA and CA extracts. The results indicated that CA could reduce scar formation and improve the wound healing process. Water retention rate, weight loss rate, antibacterial activity, and in vitro cytotoxicity were tested as well. The results indicated that skin fibroblast cell viability was accelerated and the water retention of the patches was improved, which contributed to the exudate absorption [84].

### 2.3. Foams

Foams are another group of polymer wound dressings that have been previously used with additives to accelerate wound recovery. There are some reports on using plant-derived extracts as antibacterial agents in foam-based dressings. Nantaporn et al. prepared polyurethane foam sheets containing silver and asiaticoside (AS) (an extract derived from *Centella asiatica* plant) for healing dermal wounds. AS in a foam formulation played an essential role to increase the healing rate. The MIC of the additives against *P. aeruginosa*, *S. aureus*, *E. coli,* and *B. subtilis* were in a range of 0.4–3.1 ppm. However, the foam dressing released 4–5 ppm of the additive. The clear zones from disc diffusion assays were statistically larger than other tested formulations [21]. AS has been proved to be efficiently mixed with other polymers in different studies. Phaechamud et al. developed an absorbent chitosan-based dressing containing silver and asiaticoside as an additive. This dressing showed a successful controlled drug release along with angiogenic activity, indicating the potential to be further utilised as absorbents in medical wound dressings [85]. In what follows, the scheme of the preparation technique has been demonstrated in Figure 4.

The other group of natural plant-based antibacterial additives used in foams is essential oils. The antibacterial activity of plant essential oils such as oregano and thyme has been proven previously, with MIC values of 0.0781 µL/mL [86] and 0.125 mg/mL [87], respectively. Adding these oils to a natural polymer such as sweet potato starch-based foam, along with their antibacterial activity, may lead to a lower degradation under the thermoforming temperature and higher mechanical resistance [88].

### 2.4. Films

Films have previously been used as bioactive wound dressings [10]. These types of wound dressings have been used with both plant extracts and essential oils. Some studies have shown the utilisation of different plants and plant extracts in film dressings. These plants are normally chosen based on their healing and antibacterial properties. Koga et al. developed an alginate film containing *Aloe vera* (*Aloe barbadensis Miller*) gel [89]. *Aloe vera* has already exhibited several pharmaceutical activities, such as the ability to promote the healing process as well as the ability to stimulate the proliferation of fibroblasts [90]. After characterising the different aspects of films containing *Aloe vera*, the results indicated adequate transparency, uniformity, mechanical tensile strength, and hydration capacity, which makes them an ideal candidate to be used as dressings. Furthermore, the films modulated the inflammatory phase, increased angiogenesis, and stimulated collagenesis, which leads to improved healing [89]. Figure 5 demonstrates the preparation process for these types of film.

The second group of additives used in film wound dressings are essential oils. Several types of essential oils have been used as an additive to optimise the antibacterial properties of film dressings. Clove, cinnamon, chamomile, thymol, lavender, tea tree, peppermint, *Eucalyptus globulus* juvenile, lemongrass, and lemon are some of the essential oils that have been used as antibacterial additives [91,92,93,94,95].

A combination of gelatin with clove essential oil (CEO) and hydrotalcite (HT) nanoparticles was prepared by Guilherme et al. as a wound dressing. In this study, CEO-containing films exhibited bactericidal activity against *S. aureus* and *E. coli*. HT was also hypothesised to relate positively to the antimicrobial performance of using films and enhance physical properties, which was lowered by the CEO [91].

One of the challenges in preparing films containing essential oils is choosing the proper oil to be used in the process. In this context, comparisons have been made between using each type of essential oil in a wound dressing environment. Liakos et al. used various types of essential oils such as lavender, tea tree, peppermint, *Elicriso italic*, cinnamon, *Eucalyptus globulus*, lemon, and lemongrass as an additive in sodium alginate matrixes. The produced films were tested for their antibacterial and anti-fungal properties. Among all the samples tested against *E. coli*, the cinnamon essential oils showed the largest inhibition zone of 12 mm, followed by lemongrass essential oil with an inhibition zone of 3 mm. The results of the antibacterial tests along with their stability indicates that films containing essential oils have the potential to be used as antibacterial wound-dressing materials [93].

### 2.5. Dermal Patches

The drugs used in these types of wound dressings should be penetrable to the skin, which makes most drugs unsuitable in this application. Solubility and diffusivity are two factors that determine the maximum skin penetration flux [96]. Some botanical-based additives have been used to improve the characteristics of dermal patches used in skin care and the prevention of mosquito bites [97]. In a study by Sroczyk et al., a polyimide patch was loaded with blackcurrant seed oil for atopic skin hydration studies. The application of these patches was against atopic dermatitis as a common disease among children. In this disease, gamma-linoleic acid is decreased, so the blackcurrant seed oil was used to restore the gamma-linoleic acid deficiencies. Based on the results, these patches adjust to skin movements, are stable with plant oils, and exchange air due to their high permeability, which makes them a good candidate to be used in skin care and treatment [97]. The process scheme has been demonstrated in Figure 6. There are different types of botanical-based oils with a high level of gamma-linoleic acid that can be used as additives instead of blackcurrant seed oil, such as *Nigella sativa* [98], borage [99], hempseed [100], and evening primrose [101].

As previously mentioned, another application of botanical-based skin patches is in the prevention of mosquito bites. In this case, essential oils as additives in patches act as insect repellents. Chattopadhyay et al. developed a patch from an optimised mixture of cinnamon, lemongrass, and eucalyptus essential oils embedded into ethylcellulose and polyvinylpyrrolidone polymer patches. These patches were shown to be safe and effective and to contain good physico-chemical properties at room temperature. The additives in this case are not only environmentally friendly but also make the patch more effective than the previous synthetic commercial products by providing complete protection for a longer time [102].

### 2.6. Fibers and Nanofibers-Based Electrospun Polymers

Bioactive agents added during nanofiber production have been shown to improve the wound healing process [10]. There are several strategies to tailor bioactive additives into the fibres, including emulsion electrospinning, blend electrospinning, co-axial electrospinning, and surface immobilization [103].

There are several studies indicating the use of natural botanical-based bio-additives such as plant extracts and essential oils in electrospun polymer wound dressings.

Plant extracts have been added to the polymer electrospun fibres based on the final properties required for the wound dressing. Numerous types of plant extracts have been used as an additive to nanofibers such as *Azadirachta Indica* [104], tumeric [105], *Clerodendrum phlomidis* [106], *Gymnema sylvestre* [107], *Carica papaya* [108], *Aloe vera* [109], *Lawsonia inermis* [110], *Garcinia mangostana* [111], mucilage [112], clove [113], *Ataria multiflora* [114], pomegranate [115], *Achillea lyconica* [116], corn [117], fenugreek [118], henna [119], and chamomile [120].

These extracts have been proved to be effective in diabetic wound dressings. In a study by Ranjbar-Mohammadi et al., curcumin extracted from turmeric was used as an antibacterial additive in polycaprolactone electrospun fibres. The experiments indicated that the wound dressing was active for the treatment of diabetic wounds. Exhibiting an MIC of 62.5 µg/mL against *P. aeruginosa* [121], curcumin showed a more accelerated wound healing process in comparison with the blank sample [105]. Another application of nanofibers containing plant extracts is skin tissue engineering. Henna leaf extract-loaded chitosan-based nanofibrous mats were used as a wound dressing by Yousefi et al. The final product displayed efficient antibacterial activity due to *Lawsonia inermis* (Henna) leaf extracts in mats (2 wt%), with zones of inhibition against *S. aureus* and *E. coli* of 18 mm and 25 mm, respectively. The presence of henna extract caused a reduction in the fibre diameter of the mats, which makes it favourable for wound healing applications due to increasing the surface area. Furthermore, the combined advantageous features including high biocompatibility, synergistic antibacterial activity, and acceleration of wound healing can be observed by using this additive in a mixture with polymer nanofibers [119].

The next group of botanical-based additives used in nanofiber polymer wound dressings is essential oils. Different types of essential oils have previously been used as additives in a mixture with polymer nanofibers targeting wound bacteria. These plants include lavender oil [122], thyme oil [123], cinnamon oil [124], and rosemary/oregano oil [125] that have shown antibacterial activity against the most common wound bacteria such as *S. aureus*, *E. coli*, and *P. aeruginosa* [122,123,124,125,126].

An improved wound healing device using encapsulation of cerium oxide (CeO2) and peppermint oil (PM oil) on polyethylene oxide/graphene oxide (PEO/GO) electrospun polymeric mats was shown by Suganya et al. This study involved testing against Gram-positive bacteria (*S. aureus*) and Gram-negative bacteria (*E. coli*) and evaluated in vitro cytotoxicity. The results indicated that the CeO2-PM oil-PEO/GO nanofibrous mats were less toxic to the L929 fibroblast cells. Furthermore, evaluations demonstrated that the incorporation of the plant-based bioactive agent and CeO2 in a nanofibrous mat accelerates re-epithelialization and collagen deposition, which makes the system an efficient potential candidate to be applied as wound dressings with skin infections [127]. The MIC values for peppermint essential oils are 3.1 µL/mL and 6.3 µL/mL against *S. aureus* and *E. coli*, respectively [128]. In what follows, the preparation technique of CeO2-PM oil-PEO/GO nanofibrous mats is demonstrated in Figure 7.

### 2.7. Membranes

Another group of wound dressings that have been used in combination with plant-based natural additives are membranes. Both essential oils and plant extracts have shown the ability to optimise the characteristics of the final dressings. Egri et al. developed *Hypericum perforatum* oil-loaded polycaprolactone membranes to be used in wound dressing applications. After investigating the mechanical strength and antibacterial activity, the product exhibited sufficient elasticity and activity against *S. aureus* and *E. coli,* with inhibition zones of 8–13 mm and 10–12.2 mm, respectively. Not having the risk of adhering to the wound surface, not having apoptotic/necrotic effects, being biocompatible, and having a proliferative effect on cells are some of the advantageous features of the *Hyperium perforatum*-loaded membranes [129]. The preparation scheme of this membrane is demonstrated in Figure 8.

Another type of essential oil used in membranes is *Artemisia argyi.* The efficiency of this essential oil has previously been investigated against wound bacteria such as *S. aureus*, *P. aeruginosa* and *E. coli,* with MIC values of 16 µg/mL, 64 µg/mL, and 32 µg/mL, respectively [130]. Ting-Ting et al. fabricated *Artemisia argyi* oil-microcapsule (AAO-MC)/PVC fibrous membrane wound dressings and showed that the production process was enhanced using emulsification-internal gelation. The results showed excellent stability and a slow release of the oil. Furthermore, the produced membrane showed good water vapor transmission and high hydrophilicity as well as an excellent antibacterial rate of 94.3%, which is calculated by the difference between the colony counts of the blank specimen and the colony counts of culture medium that has been cultured with a bacterial solution for a specified time divided by the colony counts of the blank specimen [131].

Based on the targeted bacteria and the final characteristics, other types of essential oils may be used as additives, such as cabreuva (*Myrocarpus fastigiatus*) [132] and oregano [133]. The MIC values of pure oregano essential oil have been determined to be 0.25 mg/mL, 0.64 mg/mL, and 0.16 mg/mL against *E. coli*, *P. aeruginosa*, and *S. aureus*, respectively [134,135].

The addition of cabreuva essential oil to poly (vinyl alcohol) membranes proves its effectiveness against *S. aureus*. Its capacity to produce cell regeneration along with no detectable toxicity makes it a suitable dressing for superficial burns or minor wounds [132]. Oregano essential oils have been used with poly (L-lactide-co-caprolactone)/silk fibroin membranes as shown by Khan et al., showing a highly active membrane against both Gram-negative (*E. coli*) and Gram-positive (*S. aureus*) bacteria. The results indicated an accelerated healing process, boosted granulation, and also re-epithelialization, which confirms its potential to be used as a wound dressing [133].

### 2.8. Polymer-Drug Conjugates

Linkers used for the conjugation of drugs to polymers function to control the drug release in a pH specific manner and in the presence of enzymes depending on the chemistry of the linker employed [136]. For improving the therapeutic advantages of this type of wound dressing, moiety and solubilising units are also incorporated into polymer–drug conjugates [137,138]. Several studies indicate the use of plant extracts and essential oils conjugated with polymers. Some of the essential oils that have previously been used in polymer nanocarriers are thyme [139,140], peppermint oil [141], green tea oil [141], etc.

In a study by Shetta et al., peppermint and green tea essential oils were encapsulated into chitosan nanoparticles using the emulsification/ionic gelation method. The final product was tested against *S. aureus* and *E. coli,* showing minimum bactericidal concentration (MBC) values of 1.11 mg/mL and >2.72 mg/mL for peppermint oil and 0.57 mg/mL and 1.15 mg/mL for green tea, respectively, demonstrating their potential to be used in wound dressing applications [141]. Figure 9 demonstrates the preparation steps of this product.

Another group of botanical-based antibacterial additives with the potential to be conjugated with polymers are plant extracts. Some of the utilised plant extracts conjugated with polymer wound dressings are polyphenolics and hydrolysable tannins from *Hamamelis virginiana* [142], seaweed extract [143], *Mcrotyloma uniflorum* [144], *Aloe vera* [145], and curcumin [146].

In a study by Yang et al., gallic acid was conjugated to a 2-hydroxy (ethyl methacrylate-co-2-diethylamino) methacrylate hydrogel. Gallic acid used in this study was extracted from an Indian plant called *Terminalia bellinca,* showing antioxidant and cytoprotective characteristics. The multifunctional hydrogel was used as a carrier for cell therapy and drug delivery applications. The results indicated that the product caused a faster recovery in affected tissues, which shows their significant potential to be used in medical applications [147].

### 2.9. Other Polymer Wound Dressings

Other types of polymer wound dressings including 3D-printed scaffolds, emulgels, and nanoemulgels have been used with various plant-based antibacterial additives previously. There are several studies indicating the use of essential oils and plant extracts in these types of wound dressings.

In a study by Ilhan et al., *Satureja cuneifolia* plant extracts were blended with sodium alginate and polyethylene 3D-printed scaffolds for treating diabetic ulcers. Disc diffusion testings against *S. aureus* demonstrated that the samples containing *Satureja cuneifolia* extracts (between 0.5 to 2 wt%) have an inhibition zone of 12–13 mm, which indicates their remarkable activity against Gram-positive bacteria. However, their activity against *E. coli* was reported to be in much higher concentrations (700 µg/mL) [148].

Emulgels and nanoemulgels have been used extensively with plant extracts and essential oils as an additive. *Ocimum basilicum* extracts [149], clove oil [150], rosemary oil [151], and *piper betle* oil [152] are some of these additives.

In a study by Razdan et al., clove oil-based nanoemulgels were used as a burn wound dressing. Levofloxacin nanoemulgels were combined with clove oil and were examined in vivo against *P. aeruginosa* biofilm-infected burn wounds. The product was tested against mice and the wound closure state was observed on the 1st, 3rd, 7th, 10th, and 15th day. The results indicated a faster reduction in wound size and a complete wound closure after 15 days in comparison with the samples without the additive, which were not completely closed in that period [150].

As mentioned before, one of the ways to improve wound dressing characteristics is to include bioactive additives. The role of natural antibacterial additives in polymer wound dressing groups were summarised before. In the following, different groups of plant-based natural products, as the source of novel antibacterial additives against the most common wound bacteria (*S. aureus*, *E. coli,* and *P. aeruginosa*), are discussed [69].

## 3. Plant-Based Bio-Additives as Novel Antibacterial and Antimicrobial Additives to the Polymer Wound Dressings

Two thirds of new antibacterial therapies [69], as well as several antibacterials currently in clinical trials, are natural products. The efficacy of these products is likely the result of their evolutionary process to be bioactive, providing organisms a selective advantage in the environment. [153].

Plant-based natural resources are promising antibacterial candidates for wound treatments. There are different types of plant-based antibacterial and antimicrobial agents including plant extracts, essential oils (EOs), and endophytes [31]. In what follows, a description and rationale of choosing each of these groups is discussed. Moreover, some examples in different types of each group along with their activity against targeted wound infection bacteria are demonstrated.

### 3.1. Plant Extracts

Plant extracts have been used against specific biological targets or related diseases. There are several extraction methods including cutting, chopping, macerating, and grinding raw or dried plant material followed by adding at least one solvent. Based on the requested final product, the ratio of the extracted material amount (kg) and the used volume (L) of the solvent may be different. Some of the other factors controlling the characteristics of the final products are solvent type (alcohols, oils, or water), the solvent temperature used in the process of extraction, and the time of extraction (between 1 h to 120 h). A pharmaceutically accepted excipient such as cellulose derivatives (as diluents), gelatin (as a binder), carbohydrates (as fillers), phosphate-buffered saline (as a buffering agent), polyvinylpyrrolidone (as a dispersion enhancer), and silica (as a lubricant) can be added to the embodiment to improve its formulation properties [31].

Indigenous plants are one of the most important sources of these antibacterial additives. The valuable information about their ethnobotanical use is gained from the local population’s knowledge. This knowledge can be utilised to transform these traditional medicines into clinical applications. There are some patents to protect these discoveries based on their specificity, habitat, and composition [31].

There are many plants that have been investigated for their antibacterial activity against pathogens. Most of these plants have already been used in local folk medicine for various applications. For example, Roja et al, investigated the potential inhibitory effect of methanol leaf extracts of *Acalipha alinifolia* (AA), *Delonix elata* (DE), *Digera muricate* (DM), *Hygrophilia auriculate* (HA), *Jatropha gasipifed* (JG), *Maeua oblongifolia* (MO), *Pterocarpus santalinus* (PS), *Punica granatum* (PG), *Syzygium cumini* (SC), *Gyrocaspus americana* (GA), and *Euphorbia heterophilla* (EH) on bacterial isolates of septic wound infections. Each one of these plants has been used in local folk medicine. The results indicated that PG and SC have potential antibacterial activity against the predominant isolates from septic wounds including *P. aeruginosa*, *S. aureus, Klebsiella pneumoniae*, and *E. coli* [154].

Azizah et al. studied the antibacterial activities of *E. glabra* against *S. aureus* and *S. epidermidis*. The bioactive compounds were extracted via solvent extraction and tested against selected bacteria via screening using agar diffusion methods. The results indicated activity against both bacteria, with MIC values between 32–512 µg/mL [155]. In another study, the antibacterial constituents from the indigenous Australian medicinal plant *Eremophila duttonii* F. Muel were investigated by Joshua et al. The bioactive compounds were extracted using solvent extraction with hexane, dichloromethane, and ethanol. All the compounds showed appreciable activity against Gram-positive organisms, including *S. aureus*, *S. epidermidis*, and *Streptococcus pneumoniae* [156].

In Table 2, the common medical uses of some of these plants and the chemical class of major compounds are shown for more insight into the rationale for the utilisation of these sources.

### 3.2. Essential Oils

EO fractions are the carrier of the fragrance of plants. These secondary metabolite oils include a large number of compounds based on an isoprene structure called terpenes. Having the chemical backbone of C_10_H_16_, they exist as diterpenes, triterpenes, tetraterpenes, and hemiterpenes as well as sesquiterpenes.

Terpenoids are terpenes that contain additional functional groups. Basically, essential oils are terpenoid compounds [221]. They are synthesised from acetate units and share their origins with fatty acids. Due to their extensive branching and cyclization, they differ from fatty acids [222]. Essential oils have been extensively studied due to their inhibitory activity against pathogens [31]. Terpenes or terpenoids are active against bacteria [223], fungi [224], viruses [225], and protozoa [226]. One of the examples of this activity is triterpenoid betulinic acid, which has been shown to inhibit HIV. The mechanism of action of terpenes is not fully understood to date but it has been speculated to involve membrane disruption caused by these lipophilic compounds [222]. The processing unit of essential oils have been demonstrated in Figure 10 [227].

A skin lotion was prepared using an antibacterial essential oil containing a mixture of *Camellia japonica L*. oil with simple volatile aromatic compounds extracted from *Juniperus chinesis L.* and *Aquilaria agallocha* Lam. The steam distillation process was performed at a temperature ranging from 65 to 75 °C for 45 to 50 h [228]. The use of antibacterial essential oils caused the removal of adolescent acne and prevented skin aging. After testing the effectiveness of various essential oils based on the aforementioned substances on a cohort of 100 people, the results indicated that the essential oils were reported as moderate or high in antibacterial, antioxidant, and skin moisturization characteristics as well as acne reduction [31].

Mixing different essential oils is one of the ways to optimise the characteristics of the final product. Essential oil mixtures are able to show activity against numerous strains of bacteria (such as *E. coli*, *S. aureus*, and *Shigella*, or *Shiga* toxin-producing *E. coli*), viable but not-culturable bacteria, bacterial spores, helminth, protozoan, fungus, or virus. [31].

Another technique that can be used for achieving more efficient antibacterial activity in essential oils is nanoemulsification. Lida et al. prepared five different nano-emulsions from *Lavandula angustifolia, Rosmarinus officinalis,* and *Satureja khuzistanica* essential oils (SKEO) as well as two EO constituents (carvacrol and 1,8-cineol). After characterisation, the formulations demonstrated long-term stability. The nanoemulsification of the essential oils caused a more efficient antibacterial activity against *P. aeruginosa*. The MIC for all the crude essential oils was 64 mg/mL. However, the nano-emulsion compound of *Satureja khuzistanica* and carvacrol showed a MIC of 8 mg/mL. The MIC reported for the *Lavandula angustifolia, rosmarius officinalis*, and 1,8-cineol nano-emulsions was 16 mg/mL [229].

As mentioned in the previous section, essential oils have been widely used in wound dressings against wound pathogens. Table 3 shows examples of different antimicrobial essential oils along with their reported properties in medical applications, their activity against wound-infected bacteria, and their chemical constituents.

### 3.3. Endophytes: A Novel Source of Bioactive Compounds

Endophytes are defined as the microbes colonising the internal tissues of plants, which cause no immediate negative effects [279]. They have extensive biodiversity and are considered as a sustainable source for novel pharmaceutical applications. The discovery of endophytes dates back to 1904. However, these groups of microorganisms did not receive much attention. With the discovery of paclitaxel (Taxol) from the endophytic fungus *Taxomyces andreanea*, which has been isolated from *Taxus brevifolia* as an important anti-cancer drug, the attention changed dramatically [280]. The isolation of penicillin from *Penicillium notatum* in the 1940s by Sir Howard Florey and his team alerted the world to the significance of fungi as a novel source for bioactive compounds. Plants actively combat pathogenic attack by producing antimicrobial compounds. Screening plants for endophytic isolation has led to novel and interesting compounds [281,282]. This has subsequently directed research to consider endophytes from ethno-pharmaceutically used plants as a source of new therapeutic compounds. Due to the success of some previous medicinal drugs from microbial origins, drug discovery has been more focused on microorganisms instead of plants. Thus, it has led to the consideration of endophytic fungi as a promising rich source of natural products in the search for new drug sources [283].

Endophytes can be considered as chemical synthesisers in plants; many are responsible for synthesising bioactive compounds used as a potential source of many pharmaceutical leads. These sources have been proven to show extensive potential to be used against multi-drug resistant (MDR) microorganisms [283]. They also have been proven to be useful in novel drug discovery by the chemical diversity of their secondary metabolites, to the extent that many of them are the source of production for novel antibacterial [284,285], antiviral [286,287], antifungal [288,289], anti-inflammatory [290,291], anti-tumour [292,293], and anti-malaria [294,295] compounds. These compounds are from different chemical classes, such as alkaloids [296], terpenoids [297], flavonoids [298], phenolic compounds [299], and steroid derivatives [300]. Table 4 lists these compounds, their host plants, and their bioactivity.

Endophytes are believed to provide resistance against pathogenic attack of the host plant by producing secondary metabolites [320]. They are a great source of natural products which exhibit an extensive array of bioactivity to the extent that many of the endophytic fungi are known to produce antibacterial and antimicrobial substances. Antimicrobial metabolites are defined as the low molecular weight organic natural substances, which have been made by active microorganisms at low concentrations against other microorganisms [321]. The crude extracts from the culture broths of endophytic fungi have shown activity against pathogenic fungi, bacteria, and yeasts, cytotoxic activity on human cell lines, anti-Herpes simplex virus type 1 (anti-HSV), and malaria parasites. Different antimicrobial activities by geographically different endophytes have been studied [322]. Different natural products have been produced from endophytic fungi, such as anti-cancerous, antioxidants, antiviral, anti-insecticidal, immunosuppressant, antimicrobial, anti-malarial, and anti-mycobacterial compounds [323,324,325].

It has been reported that medicinal plants can harbor endophytes [326], which protect the host plants from infectious agents and adapts them to environmental conditions. This mechanism is enhanced by contributing to the compounds produced by the host plant that protect against biotic and abiotic stress factors [327,328,329,330,331]. Some researchers have reported that in many cases, host plant tolerance to biotic stress is related to natural products produced by endophytic fungi [332].

Different factors are responsible for the rationale for choosing the proper plant among the numerous species available. Basically, reaching a particular microbial metabolite requires a particular biotope, at both environmental and organismal levels. Plants growing in an area with great biodiversity, in a unique habitat or containing novel strategies for survival are likely to be good candidates due to their unusual biology. Thus, they are considered important for researching unusual endophytic species [333]. The second group of selected plants are the ones which are asymptomatically infected with phytopathogens. These plants are likely to have endophytes with antimicrobial features [334]. Plants with an ethnobotanical history, which have been used by Indigenous people as traditional medicines, have great potential for the discovery of novel bioactive endophytes. Endophytic *Streptomyces* isolated from an Australian medicinal plant, snakevine (*Kennadia nigriscans*), is an example of this group [335].

Many Australian native plants have a long history of being used as medicinal and culinary herbs. Some of them are even considered to be equivalent to the Mediterranean herbs. Lots of Mediterranean herbs have been investigated and their therapeutic properties have been well-documented. However, there is limited information about the use of Australian native plants in medicine [336]. Some of these plants have been investigated for their antibacterial components, such as *Eremophila glabra* [155], *Eremophila duttonii* [156], and *Eremophila alternifolia* [337].

The biological activity depends on the natural products that endophytes produce in the host plant [338]. Thus, research regarding this important source of bioactive compounds has resulted in potential drug compounds as antibacterial additives. In what follows, some studies of different applications of endophytes are discussed.

In a study by Xing et al., endophytic fungi from two types of orchids called *Dendrobium devonianum* and *Dendrobium thysiflorum* were isolated and identified. The extracted compounds using ethanol as solvent were tested against six pathogenic microbes (*Escherichia coli*, *Bacillus subtilis*, *Staphylococcus aureus*, *Candida albicans*, *Cryptococcus neoformans*, and *Aspergillus fumigatus*). The antimicrobial activity of the extracts was tested using the agar diffusion method with a concentration of 100 µg/disk. The results indicated that 10 and 11 endophytic fungi extracts originating from *Dendrobium devonianum* and *Dendrobium thysiflorum*, respectively, showed antimicrobial activity against at least one of the pathogenic bacteria listed above. Out of the fungal endophytes of both plants, *Phoma* displayed strong inhibitory activity with an inhibition zone of more than 20 mm, and *Epicoccum nigrum* isolated from *Dendrobium devonianum* showed stronger antibacterial activity than ampicillin sodium [339].

In another study, Dang et al. isolated *Trichoderma ovalisporum* endophytic fungi from *Panax notoginseng* and tested their antibacterial activity. After growing the chosen isolate in potato dextrose agar medium, samples were filtered and extracted with ethyl acetate. Finally, the crude extracted compounds were tested against *Staphylococcus aureus* and *Escherichia coli* for their antibacterial activity using disc diffusion. The results indicated a bacteria-free zone diameter of 12 mm for both strains [340]. The process of growth and extraction in endophytic fungi can be observed in Figure 11 [340].

The antimicrobial and antibacterial activities of these bioactive compounds have already been scrutinised by researchers and the same classes of chemical compounds have been used extensively in wound dressing applications in several studies. The examples below describe antibacterial compounds that have been added to wound-dressing materials. As endophytes are known to produce these compounds, it demonstrates that endophytic fungi could be an alternative and sustainable source of these valuable products.

Soares et al. developed a chitosan-based hydrogel containing flavonoids isolated from *Passiflora edulis* Sims for wound healing purposes in a diabetic rat model. The results demonstrated effective wound healing ability. In addition, the formulation could stimulate the antioxidant defence system, which positively influenced the treatment of skin lesions in diabetic rats, representing their potential use as dressings in wound treatment [341].

In a study, Azzazy et al. developed chitosan-coated PLGA nanoparticles loaded with *Peganum harmala* alkaloids for wound dressing applications. In this study, the harmala alkaloid-rich fraction loaded into PLGA nanoparticles coated with chitosan in the emulsion-solvent evaporation method was used. The results indicated that the wound closure rate was superior in comparison with the blank sample. In addition, the developed formulation demonstrated synergistic antibacterial and wound healing properties, leading to efficient wound management [342].

## 4. Conclusions

Antibacterial agents derived from natural products have made a considerable impact in the development of novel materials for the treatment of wounds. Plant-based compounds, including saponins, tannins, alkaloids, alkenyl phenols, glycoalkaloids, flavonoids, sesquiterpenes, lactones, terpenoids, and phorbol esters, have contributed a large portion of these antibacterial agents. Plant extracts and essential oils are reported in numerous studies as two major sources of antibacterial additives in all types of wound dressings. In this paper, we introduced an alternative promising source of antibacterial compounds, namely endophytes, which are recognised as sources of compounds with useful pharmaceutical properties. The diverse bioactive compounds extracted from endophytic fungi with antibacterial activities should be the focus of future development as a sustainable source of chemicals for wound dressing applications. To our knowledge, there is no study showing the use of the antibacterial compounds sourced from endophytic fungi in wound dressing applications. However, the abovementioned features and characteristics of bioactive compounds existing in endophytic fungi, along with the proven antibacterial characteristics of the extracts from endophytic fungi, show the great potential of using endophytes as new antibacterial additives for wound dressing applications, leading to new and effective products to combat acute and chronic wound infections. However, there are some limitations in using endophytic fungi extracts as novel antibacterial agents, including the low concentration of the active compounds in the extraction method and the lack of adequate in vivo trials. Overcoming these limitations requires further research in developing the previous methods of extraction, designing a method for purifying active extracted compounds, and in vivo studies in order to examine the products in a practical environment, which will potentially be the future steps of scrutinising these novel antibacterial agents.

## Figures and Tables

**Figure 1 pharmaceutics-15-00644-f001:**
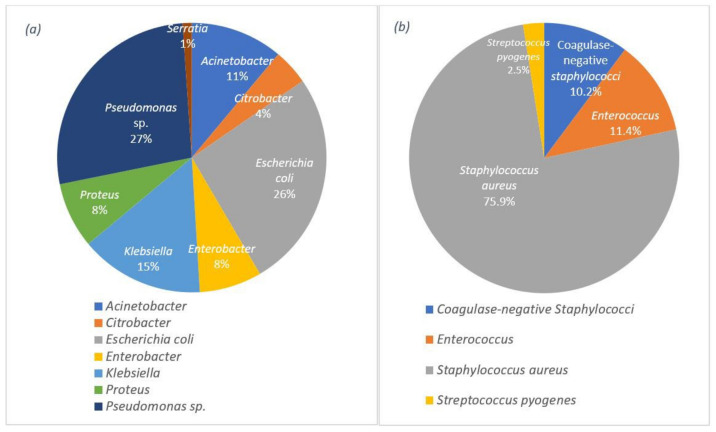
Distribution of the wound infections by: (**a**) the Gram-negative bacteria and (**b**) the Gram-positive bacteria.

**Figure 2 pharmaceutics-15-00644-f002:**
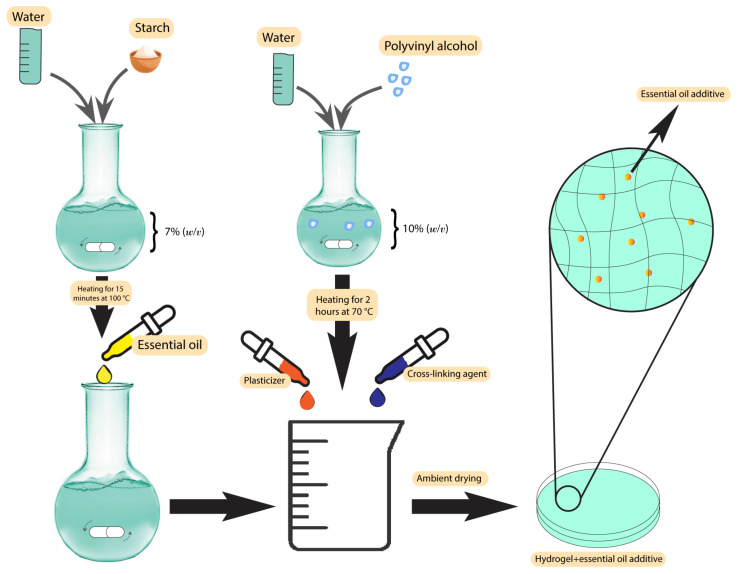
Synthesis scheme of polyvinyl alcohol/starch hydrogel membranes.

**Figure 3 pharmaceutics-15-00644-f003:**
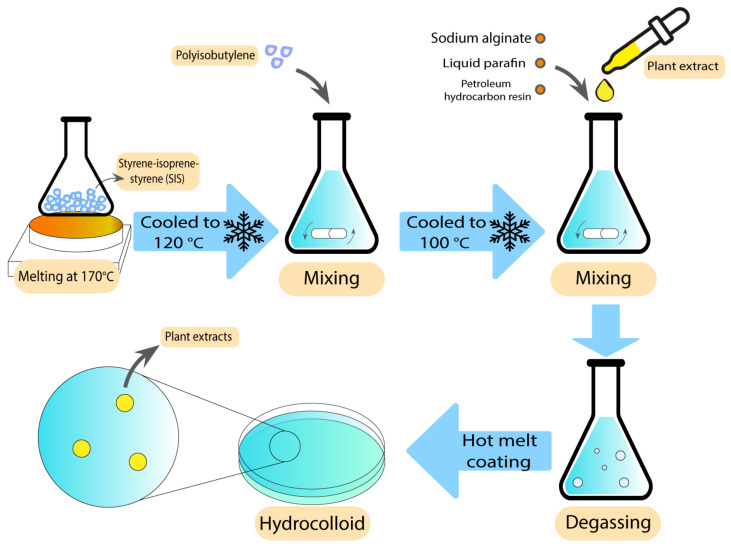
Preparation technique of alginate hydrocolloids using hot melt coating.

**Figure 4 pharmaceutics-15-00644-f004:**
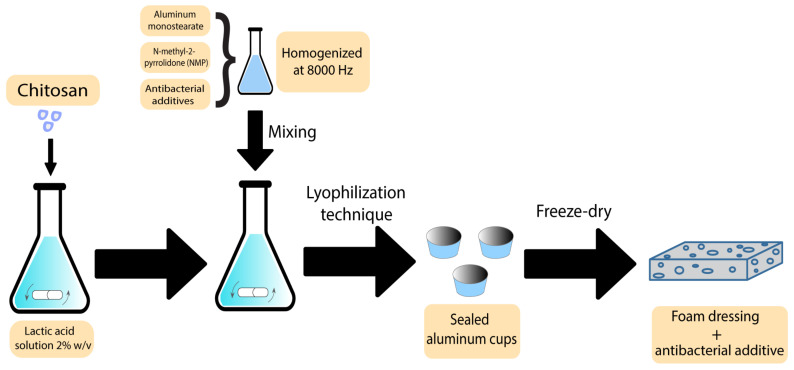
Preparation scheme of chitosan-based bioactive foams.

**Figure 5 pharmaceutics-15-00644-f005:**
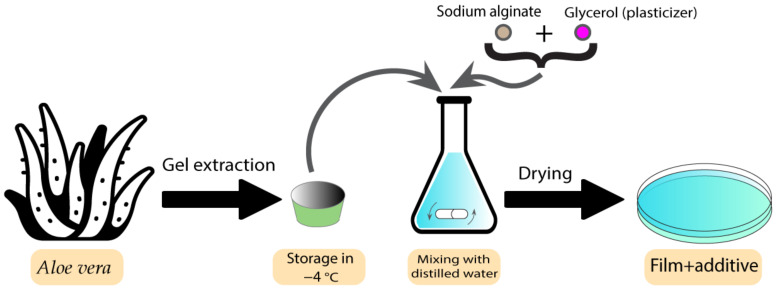
Preparation process of *Aloe vera*-containing alginate films.

**Figure 6 pharmaceutics-15-00644-f006:**
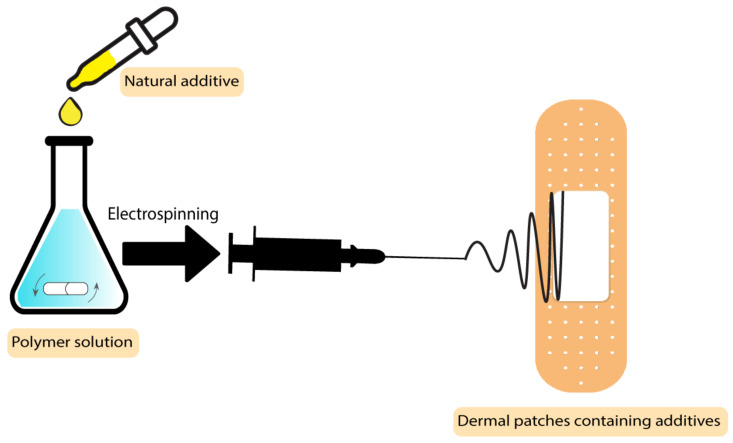
Process scheme of dermal patches containing natural additives.

**Figure 7 pharmaceutics-15-00644-f007:**
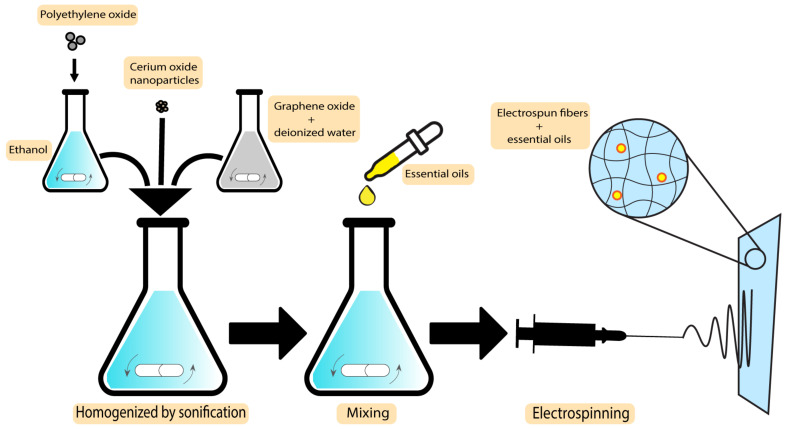
Preparation technique of CeO2-PM oil-PEO/GO nanofibrous mats.

**Figure 8 pharmaceutics-15-00644-f008:**
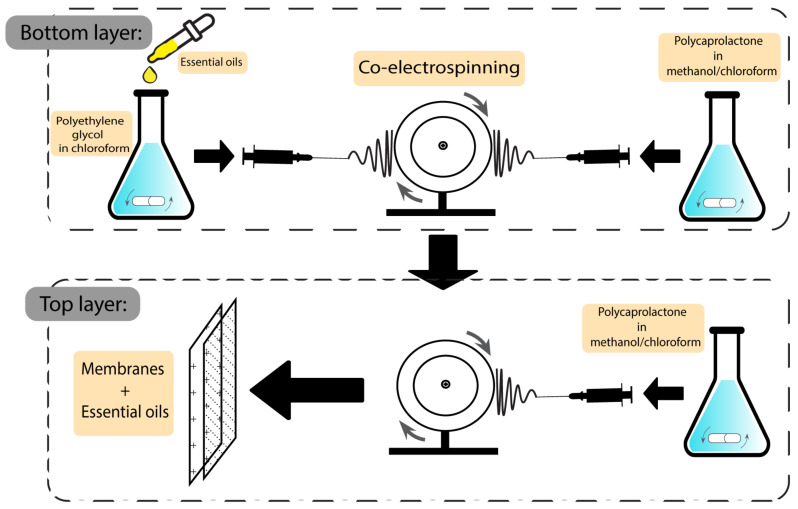
Preparation scheme of essential oil-based polycaprolactone membranes.

**Figure 9 pharmaceutics-15-00644-f009:**
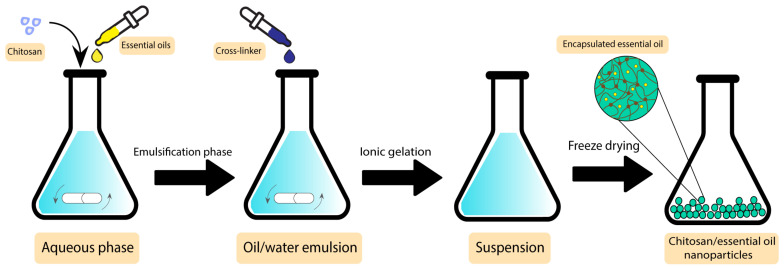
Preparation steps of chitosan/essential oil nanoparticles.

**Figure 10 pharmaceutics-15-00644-f010:**
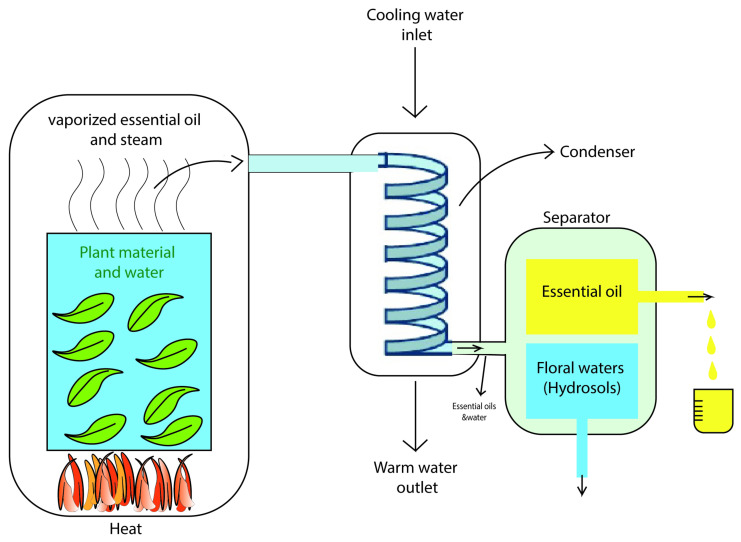
Essential oils processing units.

**Figure 11 pharmaceutics-15-00644-f011:**
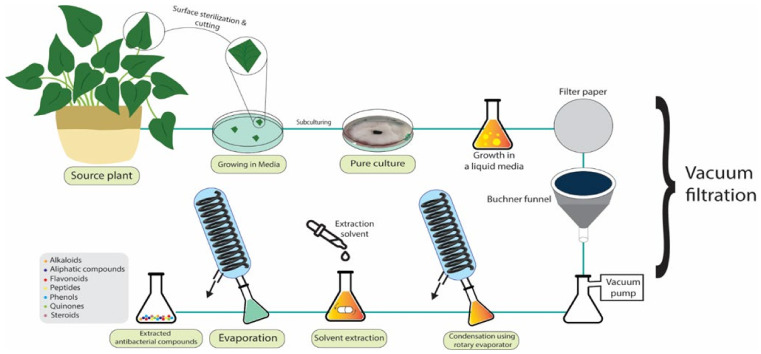
Extraction process of antibacterial compounds from endophytes.

**Table 1 pharmaceutics-15-00644-t001:** Different types of wound dressings, their wound target, and polymer type.

Variety	Description	Advantages	Disadvantages	Wound Type Application	Polymer	Ref.
Hydrogels	Water-absorbent cross-linked polymeric networks resulting from the reaction of monomers	Efficient flexibility, good ability in swelling and sustaining a significant amount of water, moisturizing, removal of necrotic tissue, good porosity, and monitoring the wound without removing the dressing	Inability to absorb enough exudates leading to bacterial proliferation, and low mechanical strength	Chemotherapy peels	Polyethylene oxide, polyvinyl pyrrolidine, Polyvinyl alcohol	[10,11,12,13,14]
				Ulcers		
				Laser resurfacing		
				Average thickness wounds		
				Donor graft sites and artificial organ wounds		
Hydrocolloids	Colloidal material (gel) constituted with elastomers and adhesives in the form of films or sheets	Excellent exudate absorption properties, transparency, enhanced angiogenesis, and formation of granulation tissue	Not permeable to gas, vapor, water, and bacteria, their debriding capability, skin maceration, and producing a foul smell	Chronic ulcers	Pectin, carboxymethylcellulose, gelatin, and cellulose	[10,13,15,16]
				Burns		
				Average thickness wounds		
				Donor graft sites		
Foams	A porous structure using capillary action as its mechanism to absorb fluids	Exudate absorbance, preventing bacteria invasion, maintaining sufficient moisture at the wound surface, being removed easily, protecting the skin around the wound, maintaining an efficient temperature, mechanical protection, being nontoxic, being cost-effective with a long shelf life	Drying out the wound in case of minimal or no exudate presence and maceration of the surrounding skin in case of exudate saturation in dressing	Chronic wounds	Polyurethane, silicone, silk fibroin	[13,17,18,19,20,21]
				Burns		
				Mohs surgery and wounds		
				Laser resurfacing wounds		
Films	Consists of adhesives, porous, and thin transparent polymers	The possibility of having a high mechanical strength, high water transmission rate, protecting the wound against bacterial infection	The possibility of having a low mechanical strength	Superficial wounds	Soy protein isolates, chitosan, polyvinyl alcohol	[13,22,23,24]
				Laser wounds		
				Surgery defect sites		
				Skin tears		
Dermal patches	Dressings consisted of a multilayered structure with an impermeable excipient-loaded film, drugs, and a release liner	Suitable for skin adhesion, not having a liquid reservoir, controlling the drug delivery rate	Needing flux moderation in case of loading with highly soluble drugs, and decrease in drug release rate with wear time, not suitable for most of the drugs	Hypertension	Poly(vinyl pyrrolidones), poly(vinyl alcohol)	[25,26,27,28,29,30]
				Topical wounds		
Fibers and nanofibers	Polymeric fibers produced with electrospinning process	Excellent mechanical properties, thermal stability, antimicrobial activity, biodegradability, control in water vapor transmission rate, oxygen permeability, fluid drainage ability, high porosity, and high surface area	Higher cost of production in some cases, hard to produce fibers with diameters less than 10 nm	Partial thickness burns	Polyurethane, collagen, silk fibroin, polycaprolactone, poly (lactic-co-glycolic acid), polyethylene oxide, etc.	[31,32,33,34,35,36,37,38,39,40]
				Diabetic ulcers		
				Bone bleeding		
				Chronic infected wounds		
				Acute wounds		
				Venous ulcers		
				Pressure ulcers		
Membranes	A thin semi-permeable barrier	Porous structure, transparency, excessive loss of water, the ability to contain an occlusive layer to impede microbial invasion	Cytotoxicity in some cases	Superficial wounds	Pectin, collagen, chitosan, chitin, alginate, zein, polycaprolactone, polyvinyl acetate, polyvinyl alcohol, polytetrafluoroethylene, cellulose, etc.	[41,42,43,44,45,46,47,48,49,50,51,52]
				Frictional wounds		
				Skin-scratching wounds		
				Skin donor sites		
				Skin with external contamination		
Polymer-drug conjugates	Polymer-based water-soluble nanocarriers conjugated with bioactive agents	Improving the water solubility of the hydrophobic drugs, enhancing the pharmacokinetic profile of the conjugated drug, extending the volume of distribution, and protecting the conjugated drug against degradation	Limitations to be applied on a large scale, low stability in vivo, short half-life, and immunogenicity	Diabetic wounds such as venous leg and lower limb ulcers	N-(2-hydroxypropyl) methacrylamide copolymer, polyglutamic acid, Poly(ethylene glycol), Polyamidoamine, hyaluronic acid, poly (vinyl ether-co-maleic anhydride), poly (vinyl pyrrolidone), etc.	[53,54,55,56,57,58,59,60,61]

**Table 2 pharmaceutics-15-00644-t002:** Plant species and their activity in wound dressing.

Scientific Name	Classes of Chemical Compounds	Common Medical Uses	Solvents Used for Extraction	Activity against Infected Wound Bacteria	References
*Acalipha alinifolia/fruticosa*	Phenolic compounds, cardiac glycosides, tannins, flavonoids, and phytosterols	Anti-bacterial, antifungal, antioxidant, and anthelmintic properties	Aqueous, acetone, and methanol	*S. aureus*	[157,158,159,160]
		asthma, pneumonia, scabies, and skin diseases		*P. aeruginosa*	
*Delonix elata*	Saponins, tannins, flavonoids, and steroids	anti-inflammatory, anti-arthritic, and antioxidant	Aqueous, ethanol, chloroform, acetone, petroleum ether, and methanol	*S. aureus*	[161,162,163,164,165]
				*E. coli*	
				*P. aeruginosa*	
*Delonix regia*	Flavonoids, alkaloids, terpenoids, steroids, and phenolic acids	Anti-diarrhoeal, anti-inflammatory, antidiabetic, antioxidant, hepatoprotective, antimicrobial, anthelmintic, wound healing, gastroprotective	Aqueous, Methanol	*E. coli*	[166,167,168,169,170,171,172,173,174,175,176,177,178,179]
				*P. aeruginosa*	
				*S. aureus*	
				*Klebsiella pneumoniae*	
*Digera muricata*	Phenol, flavonoids, alkaloids, terpenes, sterols, tannins, glycosides, and lignins	Antibacterial, antifungal, diuretic, laxative, free radical scavenger activity, anthelmintic	Petroleum ether, chloroform, ethanol, distilled water	*E. coli*	[180,181,182]
				*S. aureus*	
*Hygrophilia auriculata*	Alkaloids, terpenoids, tannins, flavonoids, and fatty acids	Medicinal usage in Indian Ayurveda	Distilled water, 50% aqueous ethanol, methanol, petroleum ether, chloroform, diethyl ether	*P. aeruginosa*	[183,184,185,186,187,188,189,190,191,192]
				*S. aureus*	
				*E. coli*	
				*K. pneumoniae*	
*Maerua oblongifolia*	Alkaloids, terpenoids carbohydrates, glycosides, phytosterols, saponins, proteins, and amino acids	Wound healing activity, treating toothache, the roots of this plant possess alternative, tonic, and medicinal properties	Petrol-Et_2_-MeOH (1:1:1), Dichloromethane/methanol, aqueous	*E. coli*	[193,194,195,196]
				*S. aureus*	
				*K. pneumoniae*	
*Pterocarpus santalinus*	Alkaloids, phenols, saponins, glycosides, flavonoids, triterpenoids, sterols, and tannins	Antipyretic, anti-inflammatory, anthelmintic, tonic, haemorrhage, dysentery, aphrodisiac, anti-hyperglycaemic and diaphoretic	70% methanol	*S. aureus*	[197,198,199]
				*P. aeruginosa*	
				*E. coli*	
*Syzygium cumini*	Flavonoids, glucoside derivatives, and phenols	Diabetes, sores and ulcers, leucorrhoea, and antidote in opium poisoning	Methanol, aqueous	*P. aeruginosa*	[200,201,202,203]
				*E. coli*	
				*S. aureus*	
*Gyrocarpus americanus*	Alkaloids	Unknown medicinal values	Ethanol, methanol, water	NA	[204,205,206]
*Punica granatum*	Polyphenols, sterols, triterpenoids, flavonoids, fatty acids, and tannins	Anti-inflammatory, anti-cancer, antioxidant, and antibacterial activity	Methanol, petroleum ether, chloroform, aqueous, chloramphenicol	*S. aureus*	[207,208,209,210,211,212,213]
				*E. coli*	
				*K. pneumoniae*	
*Euphorbia heterophilla*	Flavonoids, saponins, diterpenes, and phorbol esters	Wound healing activity, used for the treatment of constipation, bronchitis, and asthma, anti-inflammatory activity	Aqueous, petroleum ether, Butanol, ethanol	*S. aureus*	[214,215,216,217,218,219,220]
				*E. coli*	
				*K. pneumoniae*	
				*P. aeruginosa*	

**Table 3 pharmaceutics-15-00644-t003:** Application and activity of antimicrobial essential oils.

Plant	Chemical Constituents	Activity and Use	Activity against Infected Wound Bacteria	References
Clove (*Syzygium aromaticum* L.)	Eugenol, acetyleugenol, thymol, cinnamaldehyde, etc.	Anti-inflammatory, antibacterial, antifungal, anti-allergic, anti-carcinogenic, anti-mutagenic	*P. aeruginosa*	[230,231,232,233,234,235]
			*S. aureus*	
			*K. pneumoniae*	
Rosemary (*Rosmarinus officinalis* L.)	A-pinene, myrcene, 1,8-cineole, borneol, and camphor	Antioxidant, antimicrobial	*S. aureus*	[236,237,238]
			*K. pneumoniae*	
			*E. coli*	
Fennel (*Foeniculum vulgare* Mill.)	Trans-anethole, estragole	Hepatoprotective, antioxidant, anti-inflammatory, antidiabetic, antitumor, and acaricidal	*S. aureus*	[239,240,241,242,243,244,245,246]
			*E. coli*	
			*P. aeruginosa*	
Tea tree (*Melaleuca alternifolia*)	Terpinen-4-ol, 1,8-Cineol, α-Pinene, α-Terpineol, Sabinene,	Antibacterial, antifungal, used for skin treatment, airway treatment, oral treatment, and vaginal infections	*S. aureus*	[247,248,249]
			*E. coli*	
			*P. aeruginosa*	
Cinnamon (*Cinnamomum cassia*)	Cinnamaldehyde, (−)-α-Pinene/Ylangene, terpenes, aldehydes, etc.	Antibacterial, antioxidant, wound healing applications	*S. aureus*	[250,251,252]
			*E. coli*	
			*P. aeruginosa*	
Thyme (*Thymus vulgaris* L.)	1R-α-pinene, o-cymol, 4-carene, β-linalool, Camphor, Thymol, Carvacrol	Antioxidant, antibacterial	*S. aureus*	[253,254]
			*E. coli*	
Oregano	Linalool, Thymol, Carvacrol, Ethyl caprate, etc.	Antioxidant, antibacterial, anti-inflammatory	*S. aureus*	[253,255,256,257]
			*E. coli*	
Basil (*Ocimum basilicum* L.)	Linalool, 1,8-cineole, aromadendrene, and transcaryophyllene	Antibacterial and antioxidant	*S. aureus*	[258,259,260]
			*K. pneumoniae*	
			*E. coli*	
			*Shigella flexneri*	
Orange (*Citrus sinensis)*	Limonene, alcohol compounds, carvone, β-myrcene	Antibacterial	*E. coli*	[261,262]
			*P. aeruginosa*	
			*S. aureus*	
Peppermint (*Mentha × piperita* L.)	Menthol, Menthone, Limonene, β-pinene, α-pinene, Menthyl acetate etc.	Antibacterial, antifungal	*S. aureus*	[263,264]
			*E. coli*	
*Juniperus chinensis* L.	Bornyl acetate, sabinene, trans-sabinyl acetate, carotol, elemol	Air cleaning effect and antibacterial activity (mainly against *Propionibacterium acnes*), used for skincare and cleansing	*E. coli*	[265]
			*P. aeruginosa*	
			*S. aureus*	
Camellia oil	Triterpenes, Sesquiterpenes, tocopherols, phthalate esters, and cannabinoid	Skin-moisturizing effect, skin-soothing effect; effective against atopic or allergic skin conditions (effects due to high amount of oleic acid); prevention of skin dryness and alleviating itching (due to the gamma-linolenic acid content)	*E. coli*	[266,267]
			*Bacillus cereus*	
			*C. albicans*	
*Dendropanax* species	ɤ-elemene, tetramethyltricyclo hydrocarbons, β-Selinene and β-Zingiberene	Antioxidant and antimicrobial or antibacterial activities	*S. aureus*	[268,269]
			*Bacillus. cereus*	
*Portulaca oleracea* L.	Phenolic compounds, α-linolenic acid (ω3)	Alleviate skin irritation and allergic responses, antibacterial and anti-inflammatory effects/it is used mainly in acne care products and cosmetics for sensitive skin	*S. aureus*	[221,270]
			*Klebsiella oxytoca*	
*Houttuynia cordata* Thunb.	3-oxododecanal (Hou), 2- undecanone, pinene, camphene, myrcene, limonene	Effective in treating skin inflammation and atopic diseases	*S. aureus*	[271,272]
*Glycyrrhiza glabra* L.	α-pinene, β-pinene, octanol, γ-terpinene, stragole, isofenchon, β-caryophyllene, citronellyl acetate, caryophyllene oxide, and geranyl hexanolate	Detoxifying and anti-inflammatory effects, effective in alleviating skin diseases such as acne, atopy, eczema, urticarial	*E. coli*	[273,274]
			*S. aureus*	
*Ziziphus jujuba* Mill.	Triterpenoid acids, alkaloids, saponins, flavonoids, and their glycosides	Effective in moisturizing the skin and keeping the skin healthy	*E. coli*	[275,276]
			*S. aureus*	
			*P. aeruginosa*	
*Chamaecyparis obtuse* (Siebold & Zucc.) Endl.	δ-cadinene, α-pinene, γ-cadinene, α-cedrol, α-muurolene, γ-eudesmol, γ-muurolene, α-elemene and α-copaene	Sterilising effect (due to the high content of phytoncide)	*S. aureus*	[277,278]

**Table 4 pharmaceutics-15-00644-t004:** Bioactive compounds produced by antibacterial endophytic fungi.

Types of Endophytic Compounds	Endophyte	Host Plant	Main Bioactivity	Activity against Wound Infection Bacteria	References
Aliphatic compounds	*Chaetomium globosum*	*Ginkgo biloba*	Antifungal, antibacterial	*S. aureus*	[296,301]
	*Cladosporium* sp.	*Quercus variabilis*	Antifungal, antimicrobial	NA	[302]
	Fungal endophytes	Chinese herbs	Antimicrobial, antibacterial	*Klebsiella pneumonia*	[300,303,304,305]
				*S. aureus*	
				*P. aeruginosa*	
	*Phomopsis* sp.	*Excoecaria agallocha*	Antifungal, antimicrobial	*S. aureus*	[306]
				*E. coli*	
Alkaloids	*Acremonium zeae*	Maize	Antifungal, antibacterial	*Pseudomonas fluorescens*	[307,308]
				*Enterobacter agglomerans*	
	*Phomopsis* sp.	*Garcinia dulcis*	Antibacterial	*S. aureus*	[309,310]
Flavonoids	*Nodulisporium* sp.	*Juniperus cedre*	Antifungal, Antibacterial	NA	[298]
Peptides	*Cryptosporiopsis* sp., *Pezicula* sp.	*Pinus sylvestris* and *Fagus sylvatica*	Antifungal, antibacterial	*E. coli*	[311,312]
	*Fusarium tricinctum*	*Rhododendron tomentosum*	Antimicrobial	*S. aureus*	[313]
	*Penicillium* sp.	*Acrostichum aureurm*	Antifungal, Antibacterial	*S. aureus*	[314]
Phenols	*Alternaria* sp.	*Sonneratia alba*	Antibacterial	*S. aureus*	[315]
	*Phoma species*	*Saurauia scaberrinae*	Antibacterial	*S. aureus*	[299]
	*Penicillium* sp.	*Cerbera manghas*	Antibacterial	*S. aureus*	[299]
Quinones	*Ampelomyces* sp.	*Urospermum picroides*	Antibacterial	*S. aureus*	[316]
Steroids	*Colletotrichum* sp.	*Artemisia annua*	Antifungal, antimicrobial	*S. aureus*	[300]
				*P. aeruginosa*	
	*Nodulisporium* sp.	*Juniperus cedre*	Antifungal, Antibacterial	NA	[298]
	Fungal endophytes	*Daphnopsis americana*	Antibacterial	*S. aureus*	[317,318]
	*Periconia* sp.	*Taxus cuspidate*	Antibacterial	*S. aureus*	[297]
				*K. pneumoniae*	
	Fungal endophytes	NA	Antimicrobial	*S. aureus*	[319]
				*P. aeruginosa*	
				*E. coli*	

## Data Availability

Not applicable.

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
