# Peer review of "Recent Advances in Using Natural Antibacterial Additives in Bioactive Wound Dressings"

_pharmaceutics, 2023, doi:10.3390/pharmaceutics15020644_

Round 1

Reviewer 1 Report

The authors present a review of the uses of antibacterial additives in wound management dressings. For the development of the review, information from more than 300 articles is analyzed. The work is well introduced, a good context of the subject is presented, the knowledge gap is clearly specified and the objective of the review is described. Section 2, where the different strategies for the development of wound dressings and incorporating extracts or essential oils are presented, is quite complete and very well complemented by section 3 where active ingredients (extracts, essential oils, endophytes) are described in detail. Finally, the conclusions are well supported.

I recommend the publication of the manuscript with two suggestions.

1) Line 25. endophytic fungi are not vegetables

2) Introduce the "Future Directions" section

Author Response

Response to Reviewer 1 Comments

Dear Reviewer,

Thanks a lot for your comments and kind consideration. The comments have been revised and the errors have been fixed. In what follows, you may find the responses to your comments.

Comments:

The authors present a review of the uses of antibacterial additives in wound management dressings. For the development of the review, information from more than 300 articles is analyzed. The work is well introduced, a good context of the subject is presented, the knowledge gap is clearly specified and the objective of the review is described. Section 2, where the different strategies for the development of wound dressings and incorporating extracts or essential oils are presented, is quite complete and very well complemented by section 3 where active ingredients (extracts, essential oils, endophytes) are described in detail. Finally, the conclusions are well supported.

I recommend the publication of the manuscript with two suggestions.

  • Line 25. endophytic fungi are not vegetables

Response 1: The sentence was changed to: “however, there is another source of plant-derived antibacterial additives, i.e. those produced by symbiotic endophytic fungi, that show great potential in wound dressing applications”.

2) Introduce the "Future Directions" section

Response 2: Line 625-631 were added to the conclusion section: “Although there are some limitations in using endophytic fungi extracts as novel antibacterial agents including the low concentration of the active compounds in the extraction method and the lack of adequate in vivo trials. Overcoming these limitations requires further research in developing the previous methods of extraction, designing a method for purifying active extracted compounds, and also in vivo studies in order to examine the products in a practical environment, which will potentially be the future steps of scrutinizing these novel antibacterial agents”.

Reviewer 2 Report

1- It is preferable to add other materials as 3d printed scaffolds, emulgels and naboemulgels as antibacterial wound healing materials.

2- Please add schematic diagrams to show the preparation techniques for each type.

3- Please add schematic diagrams for the different wound dressings types.

Author Response

Dear Reviewer,

Thanks a lot for your comments and kind consideration. Please see the attachment.

Reviewer 3 Report

The article entitled "Recent advances in using antibacterial additives in bioactive wound dressings" is a review focused on the use of natural compounds as antibacterials for wound dressings, especially plant compounds and endophytic fungi. The abstract is appropriate and corresponds to the content of the review. Since the entire review is focused on products of natural origin, I would include this term in the title. For example, “Recent advances in using natural origin antibacterial additives in bioactive wound dressings”

The exposed information is based on a sufficient number of references. In reference to the structure and length of the review, I think it is adequate and contains the right amount of information. The use of English is correct throughout the entire manuscript.

Regarding specific changes, I propose:

- Table 1: I suggest changing the column titled “Characteristics related to wound dressing ap-plication” by a column of “Advantages” and another one of “Disadvantages”. In this way, it would not be necessary to put “advantages” and “disadvantages” in each cell of the table for those two columns.

- Line 64: Change “ecent” to “recent”.

- Figure 1: In the text “Pseudomonas aeruginosa” is mentioned but in Figure 1 only “Pseudomonas” appears. Species should be specified if known.

- Lines 114 and 116: “Clove” and “Synthesis” should not be capitalized.

- Lines 127 and 132: Morus alba should be in italics, and abbreviated in its second mention. S. aureus also in italics.

- Line 129: in vivo without script.

- Line 191 and 202: Eucalyptus with a capital letter.

- Table 2: in relation to the antibacterial activity of Punica granatum extracts, the cited references are very old, between 12 and 24 years old. I suggest introducing the following recent reference that may be helpful:

Álvarez-Martínez FJ, Rodríguez JC, Borrás-Rocher F, Barrajón-Catalán E, Micol V. The antimicrobial capacity of Cistus salviifolius and Punica granatum plant extracts against clinical pathogens is related to their polyphenolic composition. Sci Rep. 2021 Jan 12;11(1):588. doi: 10.1038/s41598-020-80003-y. PMID: 33436818; PMCID: PMC7803989.

- Figure 3: Change “Seperator” to “Separator”.

- Table 3: eliminate the entry of Aquilaria agallocha Lam. If you don't have a proper reference. Also delete entries that do not describe antimicrobial activity such as Salicornia herbacea L. or Arctium lappa L.

Overall, I think the article is good and should be accepted for Pharmaceuticals publication after addressing the changes proposed above.

Author Response

Dear Reviewer,

Thanks a lot for your comments and kind consideration. The comments have been revised and the errors have been fixed. In what follows, you may find the responses to your comments.

Comments:

The article entitled "Recent advances in using antibacterial additives in bioactive wound dressings" is a review focused on the use of natural compounds as antibacterials for wound dressings, especially plant compounds and endophytic fungi. The abstract is appropriate and corresponds to the content of the review. Since the entire review is focused on products of natural origin, I would include this term in the title. For example, “Recent advances in using natural origin antibacterial additives in bioactive wound dressings”

Response: The word “natural” was added to the title

The exposed information is based on a sufficient number of references. In reference to the structure and length of the review, I think it is adequate and contains the right amount of information. The use of English is correct throughout the entire manuscript.

Regarding specific changes, I propose:

  • Table 1: I suggest changing the column titled “Characteristics related to wound dressing ap-plication” by a column of “Advantages” and another one of “Disadvantages”. In this way, it would not be necessary to put “advantages” and “disadvantages” in each cell of the table for those two columns.

Response 1: The section “Characterization” was changed to two columns of “Advantages” and “Disadvantages”

  • Line 64: Change “ecent” to “recent”.

Response 2: The error was fixed

  • Figure 1: In the text “Pseudomonas aeruginosa” is mentioned but in Figure 1 only “Pseudomonas” appears. Species should be specified if known.

Response 3: in Figure 1 and also in the text (line 77) the word “Pseudomonas” was changed to “Pseudomonas sp.”

  • Lines 114 and 116: “Clove” and “Synthesis” should not be capitalized.

Response 4: The error was fixed.

  • Lines 127 and 132: Morus alba should be in italics, and abbreviated in its second mention. S. aureus also in italics.

Response 5: The error was fixed.

  • Line 129: in vivo without script.

Response 6: The error was fixed

  • Line 191 and 202: Eucalyptus with a capital letter.

Response 7: The error was fixed

8- Table 2: in relation to the antibacterial activity of Punica granatum extracts, the cited references are very old, between 12 and 24 years old. I suggest introducing the following recent reference that may be helpful:

Álvarez-Martínez FJ, Rodríguez JC, Borrás-Rocher F, Barrajón-Catalán E, Micol V. The antimicrobial capacity of Cistus salviifolius and Punica granatum plant extracts against clinical pathogens is related to their polyphenolic composition. Sci Rep. 2021 Jan 12;11(1):588. doi: 10.1038/s41598-020-80003-y. PMID: 33436818; PMCID: PMC7803989.

Response 8: The introduced reference was added to the table in the related section.

9-Figure 3: Change “Seperator” to “Separator”.

Response 9: The error was fixed

10- Table 3: eliminate the entry of Aquilaria agallocha Lam. If you don't have a proper reference. Also delete entries that do not describe antimicrobial activity such as Salicornia herbacea L. or Arctium lappa L.

Response 10: The entry of Aquilaria agallocha Lam was eliminated and the species with no antimicrobial activity were deleted

Reviewer 4 Report

In this manuscript (pharmaceutics-2182540), the authors have reviewed the recent advances in using various antibacterial agents derived from natural products in bioactive wound dressings. This review is interesting and timely. After going throughout the mansucript, I have found that this manuscript can be considered for publication after a minor revision. 

In conclusion section, the authors should discuss the challenges or limitations in using these natural derived antibacterial agents and the opportunities to use them in future research activities. It would provide the better insight to the readers for future research directions. 

Author Response

Dear Reviewer,

Thanks a lot for your comments and kind consideration. The comments have been revised and the errors have been fixed. In what follows, you may find the responses to your comments.

Comments:

The In this manuscript (pharmaceutics-2182540), the authors have reviewed the recent advances in using various antibacterial agents derived from natural products in bioactive wound dressings. This review is interesting and timely. After going throughout the mansucript, I have found that this manuscript can be considered for publication after a minor revision. 

In conclusion section, the authors should discuss the challenges or limitations in using these natural derived antibacterial agents and the opportunities to use them in future research activities. It would provide the better insight to the readers for future research directions

Response: Lines 625-631 were added to the conclusion section: “Although there are some limitations in using endophytic fungi extracts as novel antibacterial agents including the low concentration of the active compounds in the extraction method and the lack of adequate in vivo trials. Overcoming these limitations requires further research in developing the previous methods of extraction, designing a method for purifying active extracted compounds, and also in vivo studies in order to examine the products in a practical environment, which will potentially be the future steps of scrutinizing these novel antibacterial agents”.

Reviewer 5 Report

Very interesting review. The authors address an important wound care issue namely the antibacterial additives used in dressings. However, they must take into account the fact that many professionals can be involved in this important topic. It is surprising that they have not devoted more space to the description of the mechanisms relating to the healing mechanisms of wounds, whether acute or chronic. I therefore suggest adding this paragraph to the document. Obviously, given that this argument is innovative, modern data relating to the two-year period 2020-2022 must be reported for the description of this topic.

Author Response

Dear Reviewer,

Thanks a lot for your comments and kind consideration. The comments have been revised and the errors have been fixed. In what follows, you may find the responses to your comments.

Comments:

Very interesting review. The authors address an important wound care issue namely the antibacterial additives used in dressings. However, they must take into account the fact that many professionals can be involved in this important topic. It is surprising that they have not devoted more space to the description of the mechanisms relating to the healing mechanisms of wounds, whether acute or chronic. I therefore suggest adding this paragraph to the document. Obviously, given that this argument is innovative, modern data relating to the two-year period 2020-2022 must be reported for the description of this topic.

Response: Lines 39-49 were added to the introduction section. Reader has been referred to recent review papers for more scrutinization on the specific topic. The added section is as follows.

 “Wound repair mechanisms are consisted of four main phases including haemostasis, inflammation, proliferation and dermal remodelling. In the haemostasis phase, a blood clot is formed to prevent exsanguination from vascular damage. In this step platelet receptors interact with extracellular matrix proteins to promote adherence to the blood vessel wall. The second phase of wound healing is inflammation which is the primary defence against pathogenic wound invasion followed by proliferation as the third phase. In this phase activation of keratinocytes, fibroblasts, macrophages and endothelial cells will help the process of wound closure, matrix deposition, and angiogenesis. Finally, in matrix remodelling phase, a fibrin clot is deposited leading to the formation of a scar. For more information, readers can refer to review papers by Wilkinson et al. [3] and Carr et al. [4] scrutinizing various mechanisms of wound healing”.

Round 2

Reviewer 5 Report

The authors have answered correctly at my answers.